



# Establishment of an analytical model for remote sensing of typical stratocumulus cloud profiles under various precipitation and entrainment conditions

Huazhe Shang[1,2], Souichiro Hioki[2], Guillaume Penide[2], Céline Cornet[2], Husi Letu[1], Jérôme Riedi[2]

[1] State Key Laboratory of Remote Sensing Science, The Aerospace Information Research Institute, Chinese Academy of Sciences, Beijing, 100101, China
[2] UMR 8518 - LOA - Laboratoire d'Optique Atmosphérique, Univ. Lille, CNRS, Lille,  F-59000, France

*Correspondence to*: Husi Letu (husiletu@radi.ac.cn)

**Abstract.** Structural patterns of cloud effective radius (ER) and liquid water content (LWC) profiles are essential variables of cloud lifecycle and precipitation processes, while observing cloud profiles from passive remote sensing sensors remains highly challenging. Understanding whether there exist typical structural patterns of ER and LWC profiles in liquid clouds and how

they link with cloud entrainment or precipitating status is critical in developing algorithms to derive cloud profiles from passive satellite sensors. This study aims to address these questions and provide a preliminary foundation for the development of liquid cloud profile retrievals for the Multi-viewing, Multi-channel and Multi-polarization Imaging (3MI) sensor aboard the European Organization for the Exploitation of Meteorological Satellites (EUMETSAT) Polar System-Second Generation (EPS-SG) satellite, which is scheduled to be launched in 2025. Firstly, we simulate a large ensemble of stratocumulus cloud

profiles using the Colorado State University (CSU) Regional Atmospheric Modeling System (RAMS). The empirical orthogonal function (EOF) analysis is adopted to describe the shape of simulated profiles with a limited number of elemental profile variations. Our results indicate that the first three EOFs of LWC and ER profiles can explain >90% of LWC and ER profiles. The profiles are classified into four prominent patterns and all of these patterns can be simplified as triangle-shaped polylines. The frequency of these four patterns is found to relate to intensities of the cloud-top entrainment and precipitation. Based on

these analyses, we propose a simplified triangle-shape cloud profile parameterization scheme allowing to represent these main patterns of LWC and ER. This simple yet physically realistic analytical model of cloud profiles is expected to facilitate the representation of cloud properties in advanced retrieval algorithms such as those developed for the 3MI/EPS-SG.

## 1 Introduction

Stratocumulus cloud layers extend practically unbroken for tens to hundreds of kilometers and cover approximately 20% of

the low-latitude oceans and 50% of the subtropical and midlatitude oceans (Wood, 2015). The widespread stratocumulus imposes a negative radiative forcing as it modifies the reflection of shortwave solar radiation more than outgoing longwave radiation because of their low altitude and limited optical thickness (Arabas et al., 2009). The vertical profiles of cloud effective radius (ER) and liquid water content (LWC) inside the stratocumulus layer inferred from satellites are crucial to understand cloud microphysical processes and to quantify their radiative impacts on climate. For example, the cloud droplet profile

(CDP) helps to interpret when and where the transformation into raindrops starts by coalescence. In addition, the LWC profiles represent the cloud thermodynamic and dynamic structures of the cloud column(Carey et al., 2008).



Cloud profiles are often characterized by active sensors such as cloud profiling radars or scanning rain radar. The cloud profiling radar, e.g., NASA's CloudSat CPR (Stephens et al., 2002) and ESA-JAXA's EarthCARE CPR (Illingworth et al., 2015) are able to detect cloud liquid water droplets and/or ice crystals at a millimeter band W band (94 GHz) (Burns et al.,

2016). Scanning rain radars operating at X band (9.4 GHz), Ka band (35 GHz), and W band (94 GHz) are capable to characterize vertical profiles of cloud droplets and large ice crystals at centimeter wavelengths (Kollias et al., 2014). The Precipitation Radar (PR) on the Tropical Rainfall Measuring Mission (TRMM) operating at a frequency of 13.8 GHz can capture three-dimensional maps of the intensity and distribution of rain, rain type and storm depth (Shepherd et al., 2002). The cloud profiles estimated from active radars are limited to cross sections of cloud fields and the signal itself is prone to be overwhelmed by

strong returns from Earth's surface (i.e. ground clutters, Donovan and Van Lammeren (2001)). While active sensors are prone to uncertainties, estimating cloud profile from passive imaging sensors is even more challenging.

Owing to their much larger spatial coverage, modern passive sensors could significantly help improve numerical weather predictions if cloud vertical profiles could be obtained from their observations. However, the majority of current operational retrieval algorithms of cloud microphysical properties from passive imaging sensors still assume that the target cloud micro-

physical parameters are vertically homogeneous, which leads to uncertainties in derived cloud datasets (Grosvenor et al., 2018; Nakajima and King, 1990). This assumption is made for example in several algorithms using bi-spectral measurements in one absorbing and one non-absorbing channels (Nakajima and King, 1990; Platnick et al., 2017; Letu et al., 2020; Shang et al., 2019), or from the multi-angle polarized reflectance measurements that carry information of the amplitude and location of maxima along the scattering angle between 135° and 170° (Alexandrov et al., 2018; Bréon and Goloub, 1998; Shang et al.,

2019). Besides the uncertainties introduced by the 3D geometry of clouds, observation geometries and aerosol or surface contamination, a fundamental limitation remains the homogeneous layer assumption while particle growth, turbulence, drizzle or rain formation processes actually lead to diverse particle size profiles (Nakajima et al., 2010; Suzuki et al., 2010; Zhang et al., 2012; Nagao et al., 2013).

Yet, cloud vertical inhomogeneity can be directly observed through retrievals of ER performed using different channels in

the shortwave infrared. Platnick (2001, 2000) addressed the weighting function of in-cloud layers to the overall reflectance observed by MODIS at 1.6 µm, 2.1 µm and 3.7 µm, indicating that reflectance at 3.7 µm is more sensitive to the cloud top. Further investigations revealed that the discrepancy in the estimated effective radius from 1.6 µm, 2.1 µm and 3.7 µm can help to characterize the profile of in-cloud microphysical properties and link satellite retrievals to stages of cloud formation or precipitation (Nagao et al., 2013; Nakajima et al., 2010). To go beyond the simple diagnosis of multispectral discrepancy,

one has to explicitly account for and describe the vertical variability of cloud properties. In order to reconcile the retrievals performed using different spectral channels some studies assumed that the cloud ER profiles are linear or polylinear with no more than one turning point so that retrieval can be implemented by either a lookup table method (Chang and Li, 2002, 2003) or a radiative transfer-based iterative method (Kokhanovsky and Rozanov, 2012).

Past studies proposed to infer profile of cloud effective radius using ensembles of values at cloud top observed simultaneously

for clouds at different stages of their vertical growth and assuming that cloud-top properties are similar to the properties of a single cloud as it grows through the various height (Rosenfeld and Lensky, 1998; Alexandrov et al., 2020; Chen et al., 2020). Other authors proposed to observe cloud sides to retrieve values of ER at different levels and assuming the values of ER at



cloud surface are representative of particle size within the cloud (Alexandrov et al. 2020; Chen et al. 2020). In addition, several studies also employed auxiliary measurements from active sensors such as cloud radar systems to obtain coincident

constraint information about cloud profiles (Saito et al., 2019). It is evident that these proposed retrieval algorithms of cloud vertical profile from passive sensors leave many open questions in terms of assumption and the optimal combination of measurements.

The abovementioned studies are inspiring for cloud profile determination from sensors like the 3MI. The 3MI acquires up to 14 successive measurements of both the total reflected solar radiance within 12 narrow-band spectral channels (central wave-

lengths at 410, 443, 490, 555, 670, 763, 754, 865, 910, 1370, 1650 and 2130 nm) and the polarized radiance in all bands except 763, 754 and 910 nm (Fougnie et al., 2018; Marbach et al., 2013) . The multi-directional observations in 1.6 µm and 2.1 µm channels are expected to provide more in-cloud structural information. The unique sensitivities of polarization to cloud droplet size near cloud top  was shown to be insensitive to the sub-pixel cloud optical thickness heterogeneity (Cornet et al., 2018; Breon and Doutriaux-Boucher, 2005) while the multi-angle observations in the oxygen A-band offer a unique oppor-

tunity to characterize cloud geometrical extent (Merlin et al., 2016; Davis et al., 2018). The high information content of such combined observations opens a promising pathway to improve significantly cloud microphysical retrieval from multi-angle and polarization measurements in a near future.

This study is initially motivated by the development of an advanced cloud retrieval algorithm using multi-viewing, multi-polarization, and multi-wavelength measurements from the 3MI sensor to characterize vertical distribution of cloud properties.

When retrieving vertical profiles of the cloud ER and LWC from passive measurements, prior knowledge of additional cloud properties (i.e., the vertical profile of the cloud concentration nuclei, cloud geometrical extent, LWP) is needed because the problem is otherwise highly underconstrained. Different kind of assumptions can be made to restrict the parameters that are needed to represent the cloud profiles. In this regard, this study investigates the vertical profiles of liquid clouds generated from a large-eddy simulation (LES) model to propose a new and simple analytical description of cloud vertical profile that

would be suitable in remote sensing application.

According to in situ measurements obtained within stratocumulus cloud layers, most nonprecipitating cloud profiles show that the droplet size increases linearly from the cloud base to the cloud top, and some profiles show one or two turning points in the middle cloud layer (Lu et al., 2007; Miles et al., 2000; Pawlowska et al., 2006). In a certain number of cases, the droplet size are much smaller at the cloud top than at the middle cloud layer or cloud base(Wang et al., 2009) . The LWC profiles are

mostly documented as triangular with a maximum value (turning point) in the middle cloud layer, and the individual cloud nuclei concentration profiles are vertically homogeneous in the middle cloud layer(Painemal and Zuidema, 2011). For precipitating clouds, drizzle drops mean that the radius increases monotonically from the cloud top down toward the cloud base (Lu et al., 2009). Due to the difficulties with in situ measurements by airborne probes, model simulations, such as large-eddy simulation (LES) models (Van Der Dussen et al., 2015) and Lagrangian–Eulerian models (Magaritz-Ronen et al., 2016),

provide viable options to improve our understanding of cloud profiles. LES models with different levels of complexity can capture microphysical processes in response to the effects of turbulent mixing. Even though inherent difficulties exist in the measurements of cloud droplet size profiles from airborne sensors, these kinds of datasets are nonetheless considered the ground truth of the cloud droplet size distribution in simulated retrievals (Alexandrov et al., 2020).



The analysis of the airborne in-situ measurements and model output leads to better constrain the variables that characterize cloud profiles in satellite retrieval. Such analysis would also facilitate and improve current profile retrieval. Specifically, the link between cloud dynamic stages and cloud profiles remains unclear. To better understand the heterogeneity of the stratocumulus layer and to make appropriate assumptions for future cloud profile retrieval methods, this study aims to answer the following two questions:

1)What are the general features of cloud ER and LWC profiles specifically for the stratocumulus layer?

2)What is the impact of cloud top entrainment and precipitation on the cloud profiles?

In order to answer those questions, we simulate a large ensemble of stratocumulus cloud profiles using the Colorado State University (CSU) Regional Atmospheric Modeling System (RAMS). Based on a statistical analysis we investigate the typical features of their profiles and use those features to develop a simple yet physically realistic analytical model that could be used in cloud properties retrieval algorithm. Section 2 describes the cloud profiles datasets we adopted and the analysis methodology. Section 3 provides the results of the typical features of LWC and ER profiles. Section 4 presents the impact of cloud-top entrainment and precipitation on the patterns of LWC and ER profiles. Section 5 discusses an analytical model for remote sensing of typical stratocumulus cloud profiles, Section 6 concludes the salient findings of our study.

## 2    Data and methods

Our analysis is based on two steps described in the following subsections. We first simulate a large ensemble of cloud profiles using a LES model. The design of experiments is illustrated in Figure 1. We then perform a statistical analysis of those profiles in order to extract the main dominant features of ER and LWC profiles. Those features are later on analyzed in view of the precipitation and entrainment conditions.

### 2.1  RAMS simulations and cases

Taking advantage of the three-dimensional near-realistic characterization of the stratocumulus layer by an LES, the vertical variability of cloud microphysics is analyzed under different intensity of turbulence and precipitation. We use the LES capability of the Regional Atmospheric Modeling System (RAMS), which is originally developed at Colorado State University, to facilitate research into predominately mesoscale and cloud-scale atmospheric phenomena (Saleeby and Cotton, 2004; Saleeby and Van Den Heever, 2013). The RAMS provides a three-dimensional cloud field simulation with a detailed bulk microphysical scheme, allows interactive grid nesting capabilities, and supports various turbulence closures, shortwave/longwave radiation schemes, and boundary conditions (Pielke et al., 1992). The analysis is based on the nocturnal aircraft measurements obtained during the first research flight (RF01) of the second Dynamics and Chemistry of Marine Stratocumulus (DYCOMS-II), of which specifications are described by Stevens et al. (2003). This mission records a very homogeneous and extended stratocumulus layer, which is well suited for the study of dry-air entrainment at cloud top.

The simulations of the DYCOMS-II case are performed with a domain size of 20 × 20 × 5 km (200 × 200 × 100 bin points) for 3 hours. The horizontal resolution is fixed at 100 m, and the vertical bin spacing is 50 m. The initial state of the simulations is based on vertical profiles of potential temperature, moisture, and horizontal winds, that were adapted from Stevens et al.





(2003). From these initial fields, 4 additional simulations are carried out by slightly modifying the temperature profiles to check their effects on the stratocumulus field and notably on entrainment rates. In addition, one extra simulation is realized by modifying the humidity profile. In summary, these 6 simulations are as follows: case 1) 'Control' is the basic simulation

with the unmodified fields; case 2) 'Control + layer 150 m' is as 'Control' but the temperature inversion is 150 m above that of 'Control' (less brutal than 'Control'), expecting more entrainment; case 3) 'Control + layer 300 m' is as 'Control' but the temperature inversion is 300 m above that of 'Control'; case 4) 'Control - 4K' is as 'Control' but with a smaller temperature inversion, expecting more mixing; case 5) 'Control + 4K' is as 'Control' but with a stronger temperature inversion; and case 6) 'Extra' is as 'Control' but initialized using a slightly modified water vapor profile. The vertical gradient of water vapor

profile above 850 m in 'Extra' case is smaller than that in 'Control' case to entrain more humid air. The temperature profiles for these cases are presented in Figure 2(a).

## 2.2 EOF analysis

An empirical orthogonal function (EOF) analysis (or equivalently principal component analysis, PCA) is adopted in this study to seek for a limited number of elemental vertical profile variation that explains the maximum amount of variance. Profiles

from the RAMS are normalized by the cloud optical thickness so that the cloud top corresponds to 0 and cloud bottom to 1, and then, normalized profiles are interpolated onto 20 vertical layers. To simultaneously analyze LWC and ER profiles, we grafted every pair of LWC and ER profiles into one record. Considering that values of ER (µm) and its variance are generally larger than those of LWC ($g/m^3$), direct grafting of the two profiles leads to an overdependence on ER profiles. To balance the weights of two profiles, we multiplied the LWC profiles by a scale factor $f$ that is determined from the ratio of standard

deviation of debiased ER and LWC profiles as follows:

$$f = \frac{\overline{std(ER_{(i,t)} - \overline{ER_{(i)}})}}{\overline{std(LWC_{(i,t)} - \overline{LWC_{(i)}})}} \qquad (1)$$

where $LWC_{(i,t)}$ indicates the liquid water content of $i$ th profile in $t$ th layer ($1 \le t \le 20$), and $ER_{(i,t)}$ indicates the effective radius likewise. The bar over a quantity indicates the vertical mean. Then, the debiased liquid water content times the scale factor $(LWC_{(i,t)} - \overline{LWC_{(i)}})f$ and the debiased effective radius $(ER_{(i,t)} - \overline{ER_{(i)}})$ are grafted into one artificial profile

$X_{(i,j=1\dots40)}$ (Equation 2). Note that whether the ER profile is grafted below or above the LWC profile would not make a difference in the results of the analysis,

$$X_{(i,t=1\dots40)} = \begin{bmatrix} (LWC_{(i,t=1\dots20)} - \overline{LWC_{(i)}}) \times f \\ ER_{(i,t=1\dots20)} - \overline{ER_{(i)}} \end{bmatrix} \qquad (2)$$

$X_{(i,j)}$ could be expressed by the first three EOFs hereby:

$$X_{(i,t)} = w_1(i)EOF_1(t) + w_2(i)EOF_2(t) + w_3(i)EOF_3(t) \qquad (3)$$




Where $w_1(i,t)$ is the weighing factor for $EOF_1(i,t)$ (i.e., first dominant EOF), and $\overline{X}_{(i)}$ stands for the average profile of the $X_{(i,j=1\ldots40)}$. The $i^{\text{th}}$ ER and LWC profiles can be reconstructed by using Equation (3), the reconstructed LWC profiles will then need to be multiplied by the factor $1/f$.

### 2.3 Entrainment and precipitation calculation

In this study, we use the entrainment rate ($\varepsilon$) to quantify the inflow of air mass into the cloudy areas. The entrainment rate $\varepsilon$ is estimated depending on the relative humidity (RH) according to Equations (4) and (5). The parameterization is based on the observational evidence that mid-tropospheric humidity modulates tropical convection. The calculation of $\varepsilon$ is also used in the European Centre for Medium-Range Weather Forecasts (ECMWF) convection scheme (De Rooy et al., 2013):

$$\varepsilon = 1.8 \times 10^{-3}\{1.3 - RH(z)\}f_{scale} \tag{4}$$

$$f_{scale} = \{q_{sat}(z)/q_{sat}(z_{bottom})\}^3 \tag{5}$$

where $q_{sat}$ is the saturation specific humidity at level z and RH is the relative humidity. Stratified cloud entrainment is dependent on cloud depth (Dawe et al. 2013; De Rooy et al. 2013) and is reduced by an increased RH in the environment. This dependence has a large benefit in the general circulation model of ECMWF. We confirmed the nonlinear negative relation between cloud geometrical thickness (cloud optical thickness as well) and cloud-top entrainment characterized by the value

of $\varepsilon$. From Equations (2) and (3), it is predictable that smaller RH and larger $q_{sat}$ at level z than at the cloud base will yield larger $\varepsilon$. In this study, cloud profiles with ε at cloud top being smaller than the 25$^{\text{th}}$ percentile are considered as 'weak' cloud-top entrainment cases (WE), whereas profiles with ε at cloud top being larger than 75$^{\text{th}}$ percentile are considered as 'strong' cloud-top entrainment cases (SE).

Stratocumulus layers are mostly comprised of liquid water and do not produce as much precipitation as deep convective

clouds, but yield drizzle or light rain (Wood 2015). we estimate the precipitation from the integrated rainwater content (rainwater path) of each profile. The histograms in Figure 3 illustrate the density distributions for the intensities of cloud top entrainments and precipitation. In the following discussion, we define a profile as strong-precipitating (SP) when the rainwater path exceeds 75$^{\text{th}}$ percentile, and weak-precipitating (WP) when the rainwater path remains less than 25$^{\text{th}}$ percentile. As we mentioned at the beginning of this paragraph, the SP is characterized merely based on our statistics, therefore not comparable

to strong/heavy precipitation defined in surface meteorological observation.

## 3 Typical structures of LWC and ER profiles for stratocumulus

### 3.1 EOFs for the LWC profiles

Adiabatic lifting increases the LWC monotonically with increasing altitude, but other processes such as entrainment of dryer air, mixing process, and precipitation fallout influence the LWC profile. To examine the dominant vertical variation of the





LWC profiles among all sampled cloud regimes, we apply the EOF analysis to all instantaneous profiles from 6 LES runs as described in Section 2. The subplot within Figure 4(a) shows first three EOFs that explains more than 91% of the total variance. The first and the third EOFs account for 65% and 8% of variance, showing that the most significant variation of LWC profiles is monotonic change of LWC from the bottom to the top of clouds. The second EOF accounts for 18% of variance, indicating that the triangle-shaped polyline is an important structural characteristic besides the monotonic change. The EOF2 is indis-

pensable to represent profiles having a positive LWC deviation from the vertical mean LWC in the middle of a cloud together with negative deviations at cloud top and cloud base. Figure 4(a) illustrates the 2-D density distribution of the weighting factors of EOF1 and EOF2. The quartile lines in Figure 4(a) indicate that the number of outliers is limited, so a closer look of the density plot is shown in Figure 4(c) by removing outliers with weighting factors less than $1^{st}$ percentile or larger than $99^{th}$ percentile. The highest density of EOF 1 weighting factor is between 0.5 and 4, while that of EOF 2 is between 0.5 and 1. As

both weighting factors are mostly positive, the EOF1 can be interpreted as a representation of vertical growth, and the EOF2 as a representation of non-adiabatic process that modifies the profile.

Figures 4(b) and (d) are the binned reconstruction of LWC profiles according to the binned mean weighting factors in Figures 4(a) and (c). The fraction of entire sample that falls into a particular bin is labeled above each diagram. The profiles in bins that represent more than 3% of population are marked by solid lines. In either the quartile bin or the arithmetic mean bin, the

reconstructed LWC profiles exhibit two main shapes: monotonic increase and triangle-shaped polyline. For the profiles with near-zero EOF2 weighting factors, the reconstructed profiles show a monotonically increasing structure, as we see in the box accounting for 7.33% in Figure 4(b) and the box accounting for 32.45% in Figure 4(d). On the other hand, the triangle-shaped polyline becomes prevalent when EOF2 weighting factors becomes large, as we see in the box accounting for 8.29% in Figure 4(b) and the box accounting for 12.15% in Figure 4(d). These triangle-shaped polyline profiles may represent multiple cellular

circulations within cloud that would explain constant LWC values in the upper part of the cloud or may have entrainment that would explain the decreasing LWC in the upper part of the cloud. Therefore, nearly all profiles can be represented either by a monotonic increase or a triangle-shaped polyline with maxima occurring at the turning point close to the middle layer.

### 3.2 EOFs for the ER profiles

A similar analysis is repeated for the ER profiles to reveal the dominant structure among all the sampled cloud bins. As the

LWC and the ER profiles are simultaneously analyzed, the fraction of variance represented by every EOF is identical: 65% for the EOF1, 18% for the EOF2, and 8% for the EOF3. Figure 5(a) shows first three EOFs that together explain 91% of the variance. The first EOF is monotonically increasing, the second EOF curving similarly to the EOF2 of LWC, and the third EOF monotonically decreasing. As the first and third EOFs of LWC are nearly identical, the third EOF serves to adjust the vertical gradient of the ER profile, keeping the LWC profile unchanged. Like the second EOF of LWC, the second EOF of

ER can be approximated as a polyline with a turning point corresponding to a maximum positive difference from average ER at a normalized COT of 0.4. The density plots in Figures 5(a) and (c) are the same as Figures 4(a) and (c); they illustrate the 2-D density distribution of first two weighting factors with lines representing either quartile boundaries or arithmetic mean boundaries (with extremes removed).



Similarly to Figures 4(b) and (d), the reconstructed cloud ER profiles are shown in Figures 5(b) and (d). Regardless of bin
boundaries, most reconstructed ER profiles are triangle-shaped polylines. Figure 5(d) indicates that the most dominant profile
structure (32.45%) shows a monotonic ER growth from the cloud base to the cloud top. Others (14.16%, 12.15%, 10.5%)
show an explicit increase from the cloud base to the middle of the cloud then remain unchanged or decrease toward the cloud
top. Being consistent to the LWC profiles in Figure 4, most ER profiles can be represented either by a monotonic increase or
a triangle-shaped polyline with maxima occurring at the turning point close to the middle layer.

## 240   4 Impact of cloud-top entrainment or precipitation on the LWC and ER profiles

The variation of weighting factors for dominant EOFs as a function of precipitation and cloud-top entrainment intensities
indicates the response of cloud profiles to different cloud entrainment or precipitation conditions. To disentangle the impact
of precipitation and cloud-top entrainment, we divide the samples in 3-by-3 subsets according to three levels of cloud-top
entrainment and precipitation. Figure 6 shows the density plot of weighting factors for EOF1 and EOF2 for each subset. In
Figure 6(a), vertical and horizontal purple lines in each subplot are first, second, and third quartiles of weighting factors. On
the other hand, purple lines in the subplots of Figure 6(b) indicates the equidistance division between 1st percentile and 99th
percentile. Data points with a weighting factor less than 1st percentile or greater than 99th percentile are excluded from Figure
6(b).

Figure 6 shows that the population of points are influenced by both intensities of precipitation and cloud-top entrainment. For
example, Figure 6(b) demonstrates that the increase of the intensities of cloud-top entrainment for the SP cases (precipitation
greater than 75th percentile) does not only impact the location of the populated points, but also disperse the data points.
Regardless of the intensities for cloud-top entrainment, the stronger the precipitating is, the larger the weighting factors for
EOF1 are. Among WP cloud profiles, it is found that the stronger the cloud-top entrainment is, the smaller the weighting
factors for EOF2 are. Among SP cloud profiles, it is found that the stronger the cloud-top entrainment is, the more diversified
the profiles are.

In the following subsections, we focus on the fraction of profiles that falls into every box bounded by the purple lines in
Figure 6 to further investigate the variation of profile shape in response to entrainment and precipitation. In addition, we
propose a different classification based on the cloud-top slope of the LWC and ER profiles.

### 4.1 Impact of cloud-top entrainment

To further evaluate the impact of cloud-top entrainment on the LWC and ER profiles, we display the statistics of profiles for
WE and SE cases. Figure 7 shows the fractions of profiles that fall into 16 boxes bounded by purple lines in subplots of Figure
6 for WE and SE cases, corresponding to the first and last quartile of Figure 3(b). The analyses are performed in two binning
methods: the quartile bin boundaries (as in Figure 6(a)), and the arithmetic mean bin boundaries without extremes (as in
Figure 6(b)).

265   Figure 7 indicates that weighting factors for WE cases are populated in the center-bottom boxes, whereas those of SE are
populated in the boxes in the leftmost column. The stronger contribution of the EOF1 in representing WE profiles leads to the





larger vertical gradient of these profiles compared to SE profiles. For example, the box corresponding to a near linear profile with small gradient that accounts for 4.92% in Figure 7(a) receives 21.23% of samples in Figure 7(b), and the box that accounts for 4.50% in Figure 7(c) receives 31.84% of samples in Figure 7(d). In addition, WE profiles have smaller EOF2 weights, resulting in less pronounced polyline shapes than SE profiles. Examples can be found in the boxes in the top two rows corresponding to more pronounced polyline profile that account for in total 28.95% in Figure 7(a) receive 42.01% of samples in Figure 7(b), and in the boxes that account for in total 6.94% in Figure 7(c) receive 13.84% of samples in Figure 7(d).

### 4.2 Impact of precipitation

Similarly to Section 4.1, the impact of precipitation is analyzed by the fractions of profiles that fall into 16 boxes in subplots of Figure 6. Figure 8 shows the fractions of profiles that fall into 16 boxes bounded by purple lines in subplots of Figure 6 for WP and SP cases, corresponding to the first and last quartile of Figure 3(a).

Figure 8 indicates that weighting factors for WP cases are populated in the left-bottom boxes, whereas these of SP are populated in the boxes in right two columns (Figure 8(b)) or center-to-right bottom boxes (Figure 8(d)). The weaker contribution of EOF1 in representing WP profiles leads to the smaller vertical gradient of WP profiles compared to SP profiles. Examples can be seen from the boxes in the left two columns corresponding to smaller EOF1 weights that account for in total 85.16% in Figure 8(a) receive 15.99% samples in Figure 8(b) and the boxes that account for in total 94.98% samples in Figure 8(c) receive 44.26% samples in Figure 8(d). In addition, WP has smaller EOF2 weights, resulting in less pronounced triangle-shaped polyline profiles than SP cases. Examples can be found in the boxes in the top two rows corresponding to larger EOF2 weights that account for in total 26.77% in Figure 8(a) receive 59.94% samples in Figure 8(b) and the boxes that account for in total 2.37% samples in Figure 8(c) receive 20.52% samples in Figure 8(d).

### 4.3 Implications for the cloud profile retrieval of 3MI

Finally, to summarize the dominant LWC and ER profiles, we classify typical patterns of LWC and ER profiles into four classes. The classification is based merely on the above-turning point slope ($\frac{d(LWC_i - \overline{LWC_i})}{dt}, \frac{d(ER_i - \overline{ER_i})}{dt}$) for LWC and ER profiles since the below-turning point ER and LWC profiles mostly increase with altitude. As the reconstructed LWC and ER detrended profiles are given by the linear combination of three functions as shown in Equation (3), the above-turning-point slope is also a result of linear combination of above-turning point slopes for EOF 1-3:

$$\frac{d\left(LWC_i - \overline{LWC_i}\right)}{dt} = -0.06w_1(i) + 0.16w_2(i) - 0.06w_3(i), \tag{6}$$

$$\frac{d\left(ER_i - \overline{ER_i}\right)}{dt} = -0.36w_1(i) + 1.11w_2(i) + 0.60w_3(i). \tag{7}$$





The factors in Equations (6) and (7) are visually regressed slopes of the above-turning point EOF1-3 for LWC and ER respectively. Either for LWC or ER profiles, the slope can be greater or smaller than 0, indicating that LWC or ER decreasing or increasing toward cloud top. Hence 4 categories can be established according to Table 1.

In Table 2, 4 profile shapes and fractions corresponding to the classification in Table 1 are summarized. The statistics from all cloud profiles as well as 4 classes (WE, SE, WP, SP) defined by the entrainment and precipitation intensities are presented to evaluate the increase or decrease of a certain profile shape as a consequence of precipitation and cloud-top entrainment. From the turning point to the cloud top, the first pattern corresponds to an increase of both LWC and ER in the upper part of the profiles, the second pattern exhibits a decrease of LWC and ER values in the upper part of the profiles, the third pattern corresponds to a decrease of LWC along with an increase of ER and the fourth pattern is opposite to the third pattern. Compared to the statistics of all samples, WE cases strongly increase the fraction of pattern 1 and restrain the other patterns, SE cases decrease pattern 1 and increase pattern 2-4, WP cases reduce the fraction of pattern 1 and 4, while increase pattern 2 and 3, and SP cases decrease pattern 3 and increase the others.

## 5 Cloud parameterization scheme for cloud vertical profiles

Based on the above-mentioned analysis of typical LWC and ER profiles of stratocumulus, we propose a scheme to characterize the ER profiles using simplified triangle-shaped structures. This scheme aims to characterize the LWC and ER for the main patterns summarized in Table 2. Specifically in Figure 9, the cloud-top ER could be smaller, larger than or equal to the ER at the turning point. The proposed scheme accepts 8 input parameters, namely, the cloud geometrical thickness ($z_c$), the cloud optical thickness ($\tau$), the turning point normalized optical thickness ($t_m$) measured from the cloud top, ER at cloud base ($r_b$), ER at cloud top ($r_t$), ER at the turning point ($r_m$), effective variance of gamma size distributions ($v_e$) and the slope ($k$) of the Cloud Droplet Number Concentration (CDNC) profile (N). In this scheme, the ER at different level (defined by the normalized optical thickness $t$ in Figure 9 is characterized by the following equations:

$$r(t) = \begin{cases} r_m & 0 < t < t_m, \quad r_t = r_m \\ \left(\dfrac{t_0 - t}{t_0 - t_m}\right)^{\frac{1}{5}} r_m & 0 < t < t_m, \quad r_t \neq r_m \\ \left(\dfrac{t_1 - t}{t_1 - t_m}\right)^{\frac{1}{5}} r_m & t_m < t < 1 \end{cases} \tag{8}$$

where,

$$t_0 = \frac{r_t^5}{r_t^5 - r_m^5} t_m \text{ and } t_1 = \frac{r_m^5 - r_b^5 t_m}{r_m^5 - r_b^5}. \tag{9}$$

The power of 1/5 is selected to maximize the consistency to the existing adiabatic growth theory, in which LWC increases from the cloud base to cloud top linearly with increasing altitude. Equation (8) is equivalent to this assumption in terms of



Atmospheric
Chemistry
and Physics



Discussions

normalized optical thickness as long as the bulk extinction efficiency is close to 2 and effective variance of particle size
distribution ($v_e$) is constant.

Furthermore, we add an assumption that the CDNC profile is linear with normalized cloud optical thickness as described by
Equation (10). The CDNC profile can also be a constant value when $k = 0$.

$$N(t) = (1 + kt)N_0 \qquad (10)$$

where $N_0$ is the intercept of the regressed liner CDNC profile (i.e., cloud top CDNC).

With assumed profiles of ER and CDNC in Equations (8)-(10), other cloud microphysical parameters can be computed as
follows. Since $z_c$ is the integration of cloud geometric thickness with respect to cloud normalized optical thickness, we have:

$$z_c = \int_0^1 \frac{dz}{dt} dt, \qquad (11)$$

where the derivative inside the integral can be derived as:

$$\frac{dz}{dt} = -\frac{\pi}{(1 - 2v_e)(1 - v_e)\tau} \frac{1}{Q_{ext}(t)r_e^2(t)N(t)}. \qquad (12)$$

In this derivation, size distributions at every level are assumed to be a gamma distribution with a constant effective variance
($v_e$), but $Q_{ext}(t)$ does not need to be approximated as 2. Using the expression obtained in Equation 12, Equation 11 can be
rewritten as follows:

$$z_c = -\frac{\pi}{(1 - 2v_e)(1 - v_e)\tau N_0} \int_0^1 \frac{1}{(1 + kt)Q_{ext}(t)r_e^2(t)} dt, \qquad (13)$$

that is,

$$N_0 = -\frac{\pi}{(1 - 2v_e)(1 - v_e)\tau z_c} \int_0^1 \frac{1}{(1 + kt)Q_{ext}(t)r_e^2(t)} dt. \qquad (14)$$

All parameters in the right-hand side of Equation (14) can be obtained from Equation (9), Mie computation and from assump-
tions. Then, the number concentration $N(t)$ at any layer can be estimated using Eqs. (10) and (14). The layer-integrated optical
thickness ($\tau_i$), LWC ($lwc_i$) and ER ($r_i$) can be computed by Equations (15)-(17):

$$\tau_i = (t_{t,i} - t_{b,i})\tau \qquad (15)$$



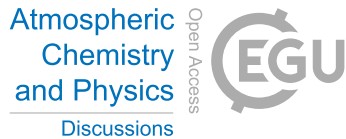

$$r_i = \frac{\int_{t_{b,i}}^{t_{t,i}} \frac{r(t)}{Q_{ext}(t)} \, dt}{\int_{t_{b,i}}^{t_{t,i}} \frac{1}{Q_{ext}(t)} \, dt} \tag{16}$$

$$lwc_i = \frac{4\pi}{3} \int_{t_{b,i}}^{t_{t,i}1} \frac{r_e(t)}{Q_{ext}(t)} dt \tag{17}$$

To demonstrate how the scheme represents the cloud profile, four profiles addressing the dominant patterns 1-4 are shown in Figure 10. For all 4 profiles, $z_c$ is fixed at 0.3 km. Profile (a) shows pattern 1 (Table 1): the scheme captures the monotonic growth of ER and LWC with a turning point at $\tau = 8.0$. The CDNC is assumed to be increasing from cloud base to cloud top with k=-0.2. Profile (b) shows pattern 2: the dominant feature is that both ER and LWC profiles above the turning point are decreasing. A constant CDNC profile from cloud base to cloud top is assumed. Profile (c) is to recreate the pattern 3 where the ER profile above the turning point continues to increase while LWC starts to decrease. Profile (d) is the opposite to Profile (c) showing ER decreasing and LWC increasing toward cloud top. In conclusion, our scheme is capable to represent the dominant patterns of ER and LWC profiles that are summarized in our EOF analysis.

Among the input parameters of the scheme, the slope of CDNC profile ($k$) is challenging to directly derive from passive measurements. We present some results of preliminary analysis to find relations between k and mean ER, cloud-top ER and the slope of ER profile ($k_{er}$ defined by $r(\tau) = k_{er}\tau + r_t$). Neither the mean ER nor cloud-top ER are found to be closely related with $k$, but $k_{er}$ and $k$ show a slight correlation. Figure 11 shows the density plots of $k_{er}$ against $k$. The parameterization of realistic $k$ is reserved for as a future work, while it appears reasonable that the $k_{er}$ and $k$ shows some correlation as they are closely related by microphysical processes in clouds.

## 6 Conclusions

Characterizing LWC and ER profiles for liquid clouds from passive satellites is challenging and always requires some level of assumptions about the cloud vertical structure to circumvent the limited information content of passive measurements. Establishing physically based constrains to facilitate the characterization of LWC and ER profiles is therefore essential to make progress towards cloud profile retrievals from passive measurements. With this goal in mind, we use simulated cloud profiles of stratocumulus from the DYCOMS-II case to analyse the main structure of LWC and ER profiles. To guarantee consistent LWC and ER structural patterns, we grafted the LWC and ER profiles when performing EOF analyses. We find that >90% of LWC and ER profiles could be approximated by monotonic increase or triangle-shaped polylines. Besides, LWC and ER profiles have similar concave and convex characteristics and similar locations of turning points. These findings suggest that it is possible to use a reduced number of parameters to describe realistic cloud profiles both in radiative transfer simulation and in actual development of cloud profile retrieval algorithms. From the first three EOFs, monotonically increasing cloud profiles with increasing LWC and ER from cloud base to cloud top are found to be the dominant profile variation, but we also observe patterns with LWC and ER nearly constant or decreasing from the turning point to cloud top. In addition, it is found that the cloud-top entrainment reduces the gradient of LWC and ER profiles whereas the precipitation increases the gradient of LWC and ER profiles. This can be explained by that the cloud-top entrainment reduces the ER and LWC at



the cloud top where they are usually larger than cloud bottom, while the precipitation further reduces the ER and LWC at cloud bottom by accretion and coalescence where they are usually smaller than cloud top.

We noticed 4 prominent patterns of LWC and ER profiles from the EOF analyses. All these patterns have monotonically increasing LWC and ER profiles in the bottom part of the clouds, while the top part of the profiles may have increasing (Pattern 1), decreasing (Pattern 2) and contradictory (Patterns 3 and 4) LWC and ER variation towards cloud top. The classification of 4 prominent patterns of LWC and ER profiles enables us to quantify the pattern variation of cloud profiles by the influence of cloud-top entrainment and precipitation. We found that the dominant patterns are Patterns 1 and 2 all the time, and they are more sensitive to cloud-top entrainment than precipitation status: WE (respectively SE) significantly increases (respectively decrease) Pattern 1 and reduce (respectively increase) the other patterns; WP reduces Patterns 1 and 4 and increase Patterns 2 and 3; and SP decrease Pattern 3 and increase the others.

Based on the analyses of cloud profiles and assumptions that the turning points of LWC and ER profiles are located at the same position in the normalized COT scale, we propose a parameterization scheme to facilitate the sensitivity studies and retrieval of cloud profiles from passive remote sensing observations, in particular from the future 3MI. In our scheme, 8 parameters are used to describe the vertical variation of cloud optical and microphysical properties. It is shown that the ER and LWC profiles can in most cases be simplified as triangle-shaped profiles with one turning point. Our tests indicate that the scheme can replicate the monotonically increasing, quasi-monotonically increasing and non-monotonically increasing cloud profiles in terms of the 4 patterns in our analyses. These results will serve as a basis to develop the retrieval of liquid cloud vertical profile from the future 3MI observations. It is expected that such retrievals will enable better description of cloud properties, in particular by providing parameters that can be more easily linked to cloud development processes of interest for nowcasting applications.

**Author contribution**

JR and CC outlined the project, HS and SH conceived of the methodology and performed the study, GP performed the simulations, HS composed the article, and all the others revised the article. JR and HL funded, supervised and encouraged the research.

**Competing interests**

The authors declare that they have no conflict of interests.

**Acknowledgement**

The authors are grateful for support from Univ. Lille, CNRS, AIRCAS and NSFC. We also thank for CSU for providing RAMS model and NCAR for the data of DYCOMS-II campaign. This work is funded by the National Natural Science Foundation of China (No. 42025504, 42175152) and Youth Innovation Promotion Association CAS (No. 2021122).



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





**Table 1. The criteria to classify 4 LWC and ER patterns according to the slope of above-turning-point profiles.**

| | $\dfrac{d\left(LWC_{(i,t)}-\overline{LWC}_{(i)}\right)}{dt}<0$ | $\dfrac{d\left(LWC_{(i,t)}-\overline{LWC}_{(i)}\right)}{dt}>0$ |
|---|---|---|
| $\dfrac{d\left(ER_{(i,t)}-\overline{ER}_{(i)}\right)}{dt}<0$ | ER: / <br> LWC: / | ER: / <br> LWC: \ |
| $\dfrac{d\left(ER_{(i,t)}-\overline{ER}_{(i)}\right)}{dt}>0$ | ER: \ <br> LWC: / | ER: \ <br> LWC: \ |



**Table 2. The main 4 LWC-ER patterns that appeared in our analyses and their percentage in terms of the following scenarios: ALL) all cloud profiles, WE) cloud profiles associated with weak cloud-top entrainment, SE) cloud profiles with strong cloud-top entrainment, WP) cloud profiles with weak precipitation, SP) cloud profiles with strong precipitation. The green and red arrows aside the numbers indicate the increase or decrease of the percentage compared with the reference statistics using all samples.**

| | [1] | [2] | [3] | [4] |
|---|---|---|---|---|
| ALL | 60.09% | 25.45% | 7.63% | 6.82% |
| WE | 81.55% ← | 11.66% ← | 4.49% ← | 2.31% ← |
| SE | 42.44% ← | 38.99% ← | 9.97% ← | 8.61% ← |
| WP | 52.20% ← | 27.81% ← | 14.65% ← | 5.34% ← |
| SP | 61.73% ← | 26.71% ← | 3.04% ← | 8.52% ← |




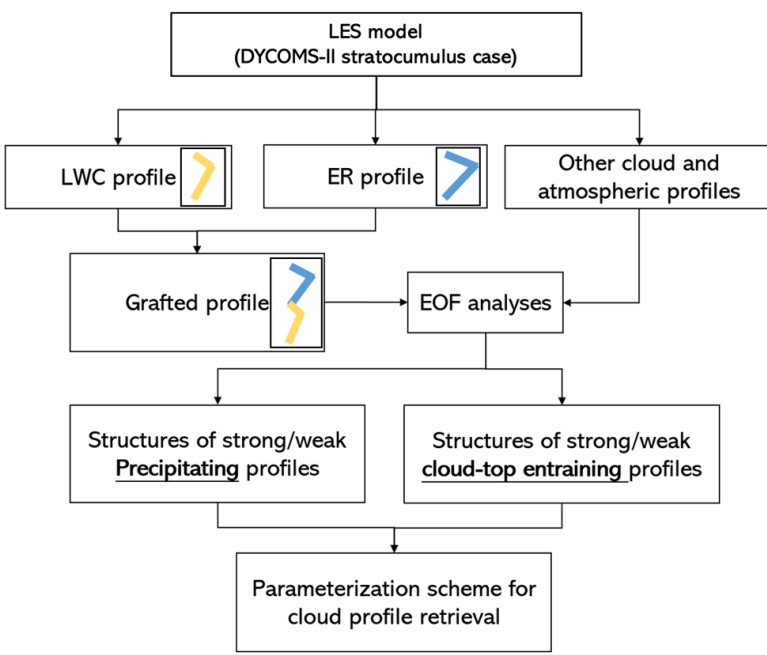

**Figure 1. the flow chart of this study**

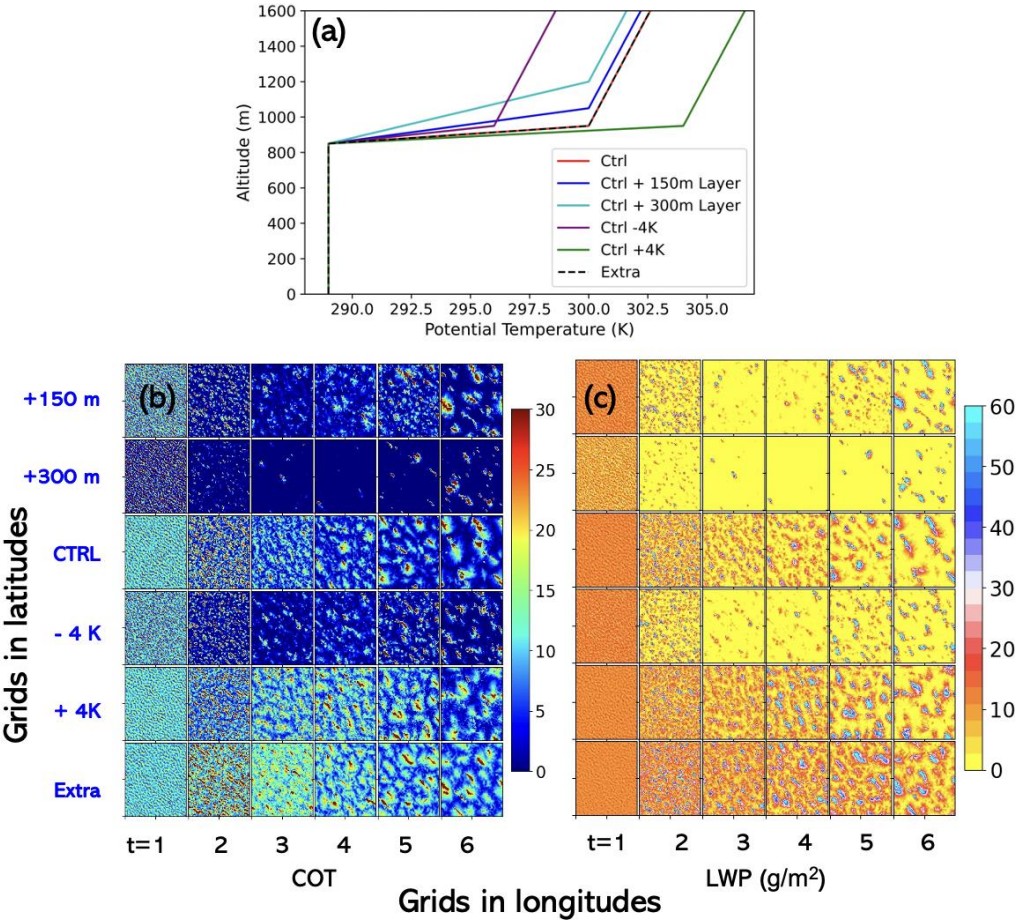

**Figure 2. a) the initial potential temperature profiles for the five cases of stratocumulus simulations: 'Control', 'Control + layer 150 m', 'Control + layer 300m', 'Control - 4K', 'Control + 4K' and 'Extra' (described in Section 2.1); b) the spatial distribution of cloud optical thickness for the 6 cases for each 30-minute timesteps, c) as in b) but for the rain water path, the cloud boundary is determined by the condensation of cloud droplets > 20 mg$^{-1}$.**





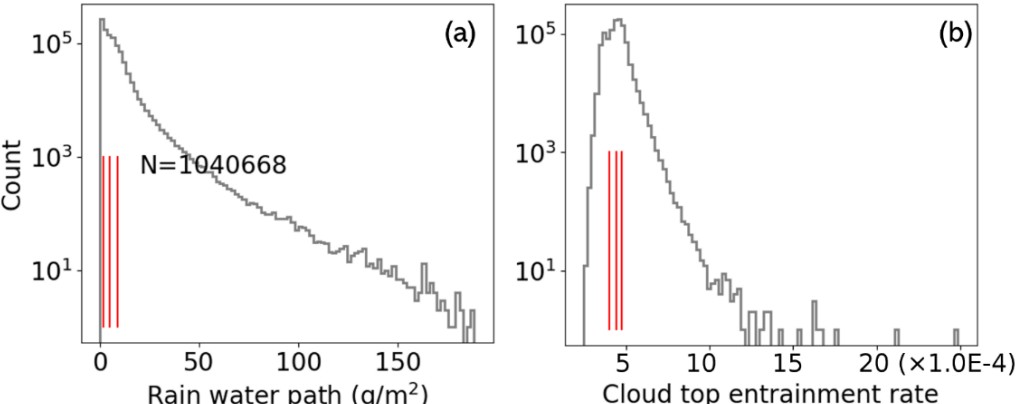

Figure 3. Histogram of the counts of the rainwater paths (a) and of the cloud top entrainment rates (b) in the RAMS cloud profiles, the red vertical lines from left to right indicate the 25th, 50th and 75th percentiles.



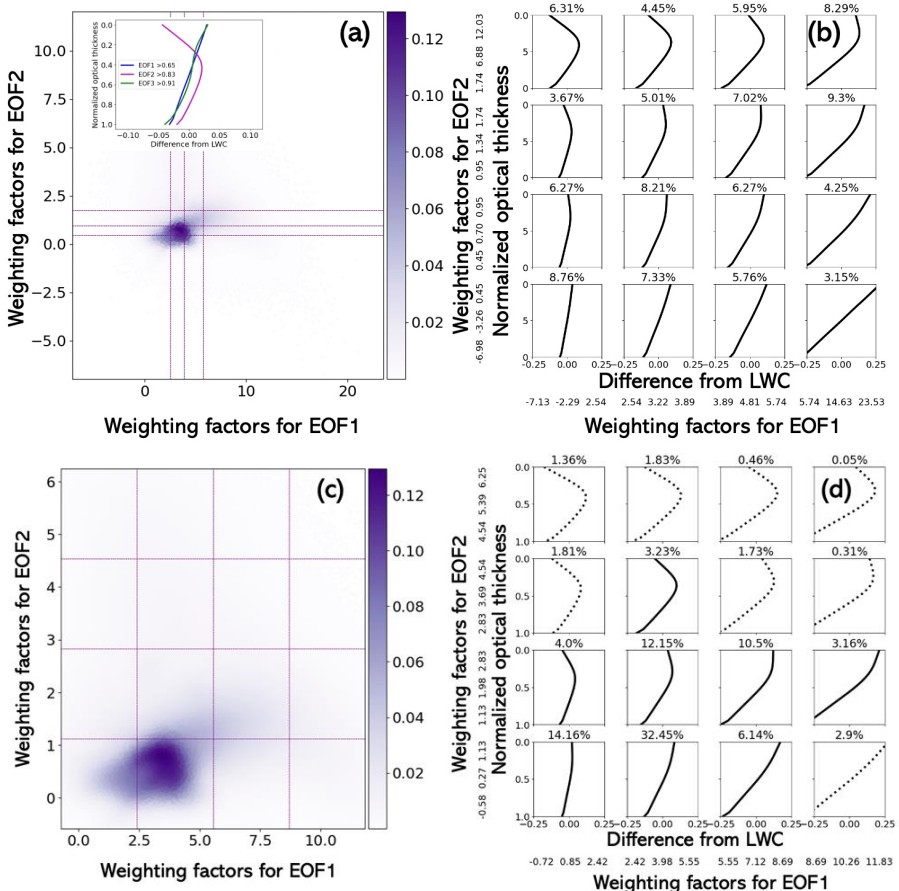

**Figure 4. (a) Density plot of EOF1 and EOF2 weighting factors for all LWC profiles. The color scale corresponds to the density of the points in percent, and the purple dotted line indicates quartile boundaries along the x and y axes. The panel inside illustrates the first three dominant EOFs. They explain the 91% of variance among all 1,040, 668 samples. (b) Cloud LWC profiles reconstructed from the EOF1-3 according to quartile bins in (a), the black dotted and solid lines denote the profiles that represent fewer than and more than 3% of the samples, respectively; (c) Same as (a) but without weighting factors exceed 99th percentile or less than 1st percentile, and with the purple dotted line indicating the arithmetic mean boundaries; (d) Same as (b) but the reconstruction is based on bins in (c).**


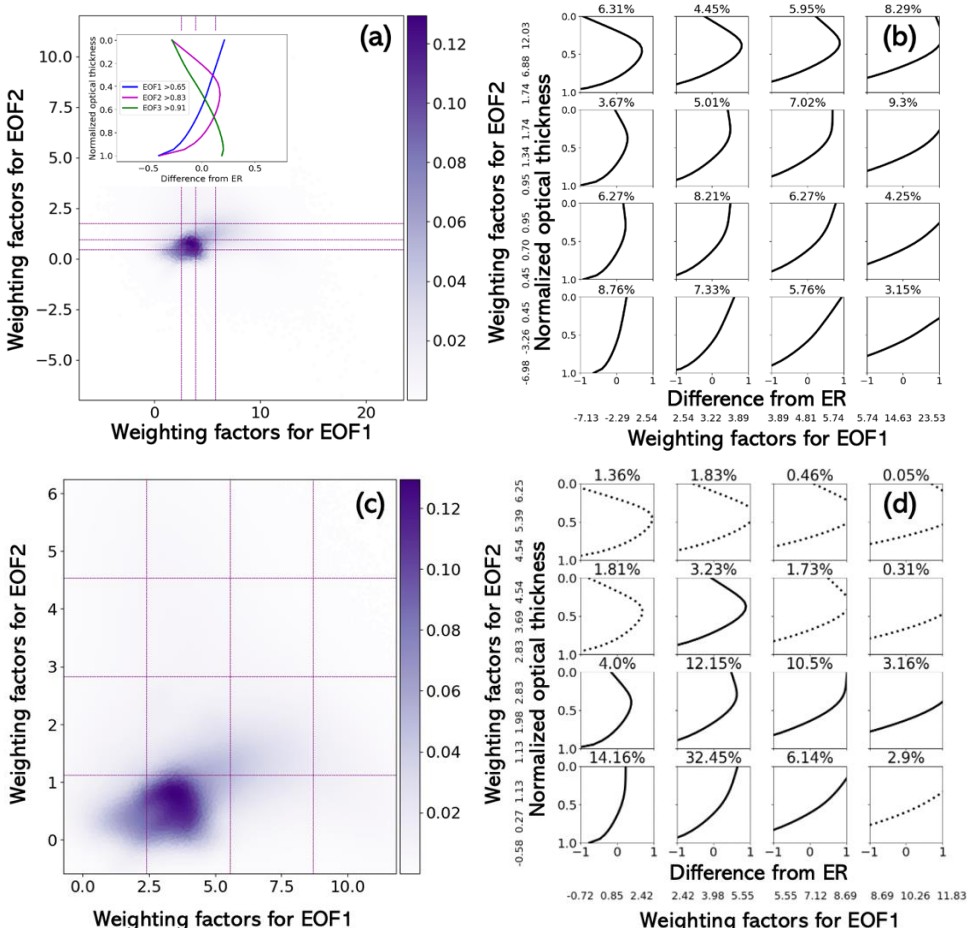

580                      **Figure 5. Same as Figure 4 but for all ER profiles.**

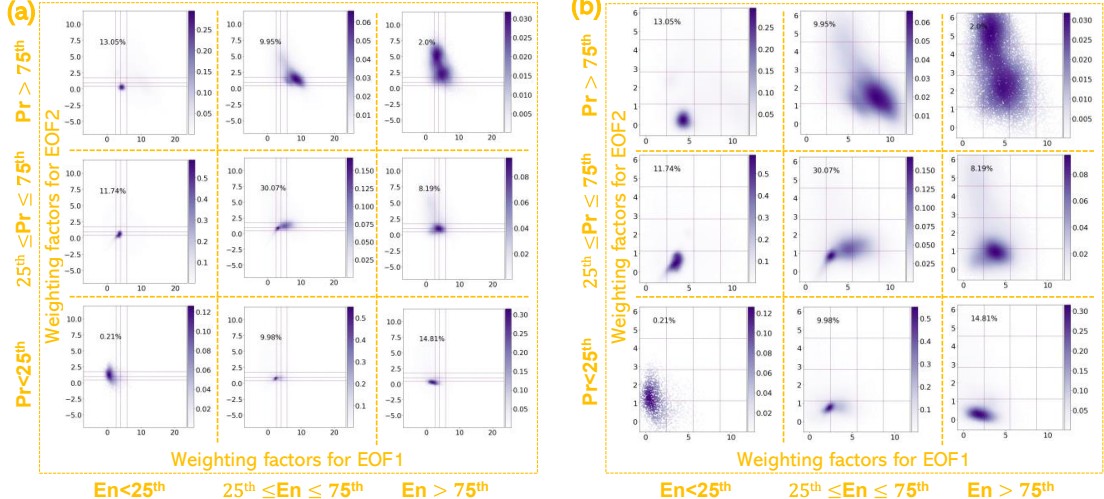

**Figure 6. The scatter plots of weighting factors for EOF1 and EOF2 for different intensity levels of cloud-top entrainment and precipitation. The weak, middle-level and strong level of cloud-top entrainment is characterized by cloud-top entrainment rate below 25th percentile, in between 25th and 75th percentiles and above 75th percentile. Similarly, three level of precipitation is characterized by the rainwater path.**



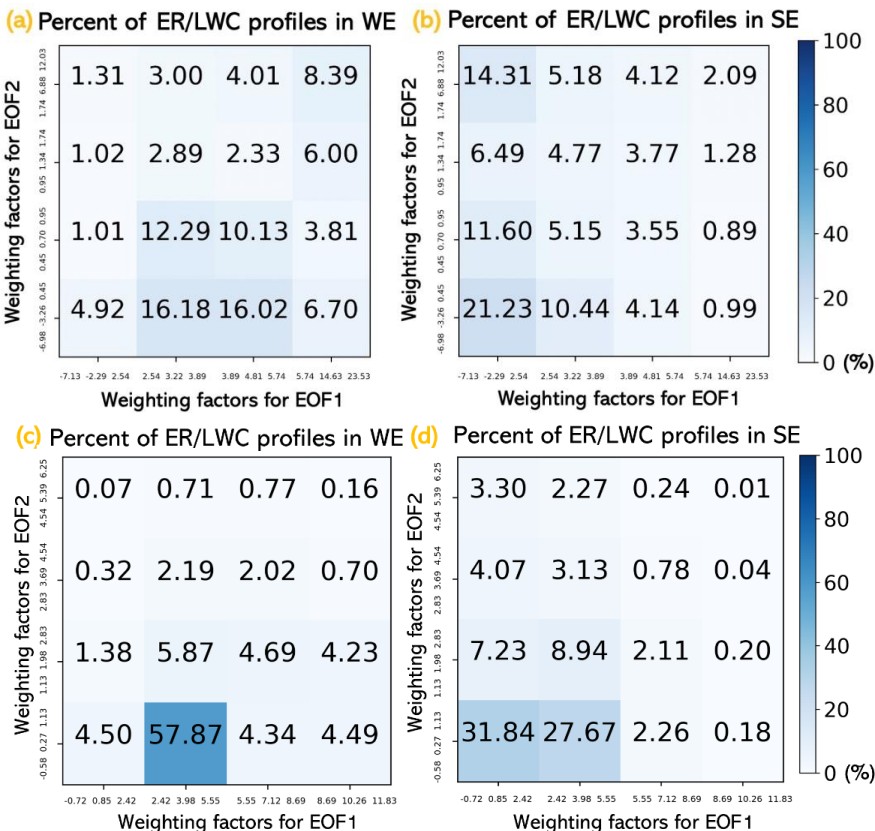

**Figure 7. (a) The percent of profiles for weak cloud-top entrainment (WE), the bins are characterized by quartile boundaries; (b) Same as (a) but for strong cloud-top entrainment (SE); (c) The percent of profiles for weak cloud-top entrainment (WE), the bins are characterized by arithmetic mean boundaries; (d) Same as (c) but for strong cloud-top entrainment (SE).**






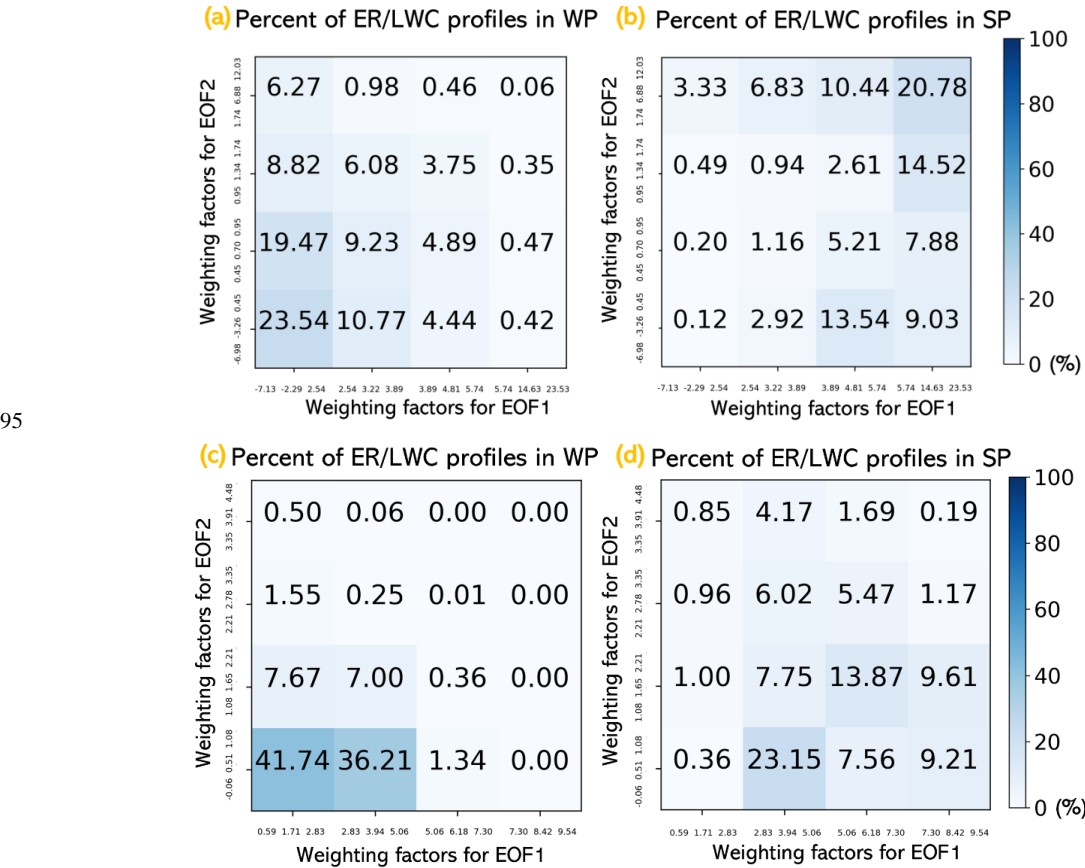

**Figure 8 (a) The percent of profiles for weak precipitation (WP), the bins are characterized by quartile boundaries; (b) Same as (a) but for strong precipitation (SP); (c) The percent of profiles for weak precipitation (WP), the bins are characterized by arithmetic mean boundaries; (d) Same as (c) but for strong precipitation (SP).**


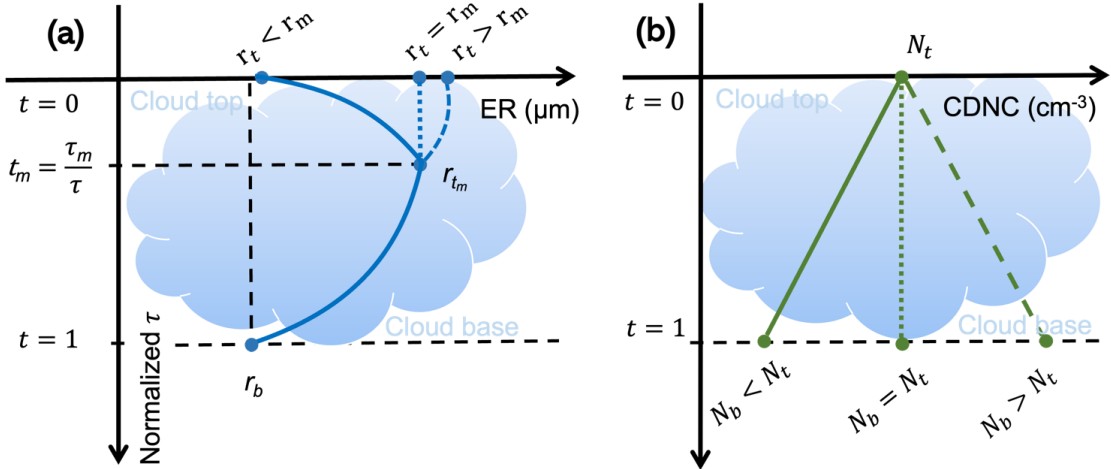

**Figure 9 (a) Simplified triangle-shaped profiles for the cloud ER and (b) simplified linear profiles for CDNC (N);Both**
$\tau$ **or normalized optical thickness (t) axes can be used to define the top (** $\tau = t = 0$ **), turning point (** $\tau = \tau_m, t = \frac{\tau_m}{\tau}$ **) and**
**bottom (** $\tau = \tau, t = 1$ **) of the ER profile.** $r_b$ **,** $r_m$ **and** $r_t$ **are the effective radii at the cloud base, the turning point and**
**cloud top, respectively. The N profile is based on linear assumption with a slope. The stratified values for the ER and**
**LWC profiles are calculated using the parameterization scheme presented in this section.**





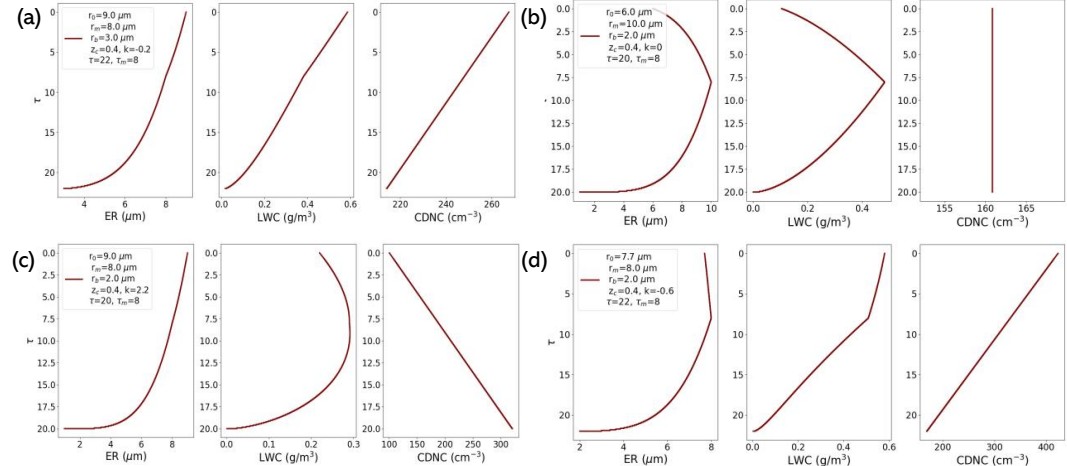

**Figure 10 Four cases of cloud ER, LWC and CDNC profiles generated by the parametrization scheme.**




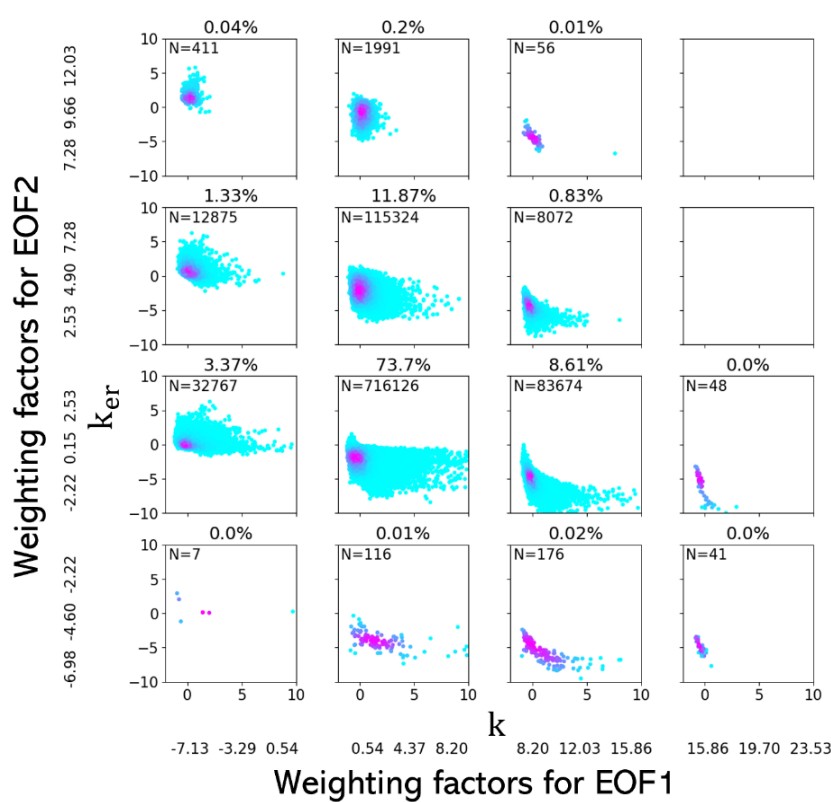

**Figure 11. The density scatter plot between the k and the that defined by $r_\tau = k_{er}\tau + r_0$.**