# Peer review of "Establishment of an analytical model for remote sensing of typical stratocumulus cloud profiles under various precipitation and entrainment conditions"

_Atmospheric Chemistry and Physics, 2022_

## Referee Comment (RC1)

[referee-annotated manuscript omitted]

---

## Author Comment (AC1)

**General comments:**

The submitted publication addresses scientific questions inherent in cloud physics and entirely fits with the scope of the ACP journal. The title properly represents the paper's contents, and the abstract is concise and complete. I appreciated the presented approach and methodology to investigate typical cloud profiles of liquid water content (LWC) and Effective radius (ER) to exploit them for satellite remote sensing. However, the paper does not bring to any substantial conclusion, and future work is needed to complete the envisioned step forward in satellite-based retrievals of the quantities mentioned above. I found the description of the dataset and the experiment quite plain and clear, thanks to fluent and precise language, and the method is reproducible. However, I have three main remarks:

1. I enjoyed reading the introduction and the multiple references presented there. However, I found that literature focusing on retrieving LWC and ER from ground-based instruments deserves a paragraph in the presented text. Besides, I also lack a more specific description of the limitations and the sources of uncertainty that affect the satellite-based estimations of LWC and ER.

2. I think that, sometimes, it is not easy to follow the interpretation that should support the results, especially regarding the conclusions extracted from the analysis of Figures 6,7, and 8. I believe that re-writing this part and providing more help to the reader to follow the presented argumentation will benefit comprehension.

3. I did not find any indication regarding the availability of the data and the code used to produce this work. I consider this a necessary condition for any publication to be accepted, so I strongly recommend that the authors make their code and dataset publicly available for open science and the reproducibility of results.

Response: Thank you very much for summarizing the main comments. The detailed responses for the main comments 1-2 are detailed in the following responses. For the main comment 3, we add a small section named "code and dataset availability" after the Acknowledgement section to clarify the data availability.

Also, we would like to briefly comment on the fact that the current paper does not bring to any substantial conclusion. To a large extent we do actually agree with this comment and fully recognized that the current work need to be completed in order to demonstrate fully how the proposed model can be used for actual remote sensing application. However, we considered that the establishment of the analytical model should be discussed separately from on-going studies about retrieval approaches because the later analyses are very much dependent upon measurements information content and retrieval strategy while the analytical model we propose in the current paper aims at being a very general basis for cloud profile retrievals, A follow on paper is under preparation to demonstrate how a specific type of measurements (namely, observations from the future Multi-channel, Multi-angle, Multi-polarization Imager) could be used to constraint the parameters of our analytical model through a Bayesian optimization. Our hope is that the currently proposed model will be considered as a basis for

the development of cloud retrieval properties using other observations and potentially other retrieval methodologies.

Please find more regarding the points above in the specific comments provided in the PDF and in the specific comments below. (Also, technical corrections in there).

I recommend publishing the paper after the minor revisions requested to solve the remarks mentioned earlier.

**Specific comments:**

Line 40: This study refers to ground-based observations, while all the others in the paragraph deal with satellite observations. I would remove this study from here and devote a separate paragraph to present a summary of the research done using ground-based obs for detecting LWC and effective radius

Line 65: It might be worth also mentioning some studies investigating LWC and effective radius profiles using ground-based observations. (for example

Roebeling, R. A., Placidi, S., Donovan, D. P., Russchenberg, H. W. J., and Feijt, A. J. (2008), Validation of liquid cloud property retrievals from SEVIRI using ground-based observations, Geophys. Res. Lett., 35, L05814, doi:10.1029/2007GL032115.

Rémillard, J., Kollias, P., and Szyrmer, W.: Radar-radiometer retrievals of cloud number concentration and dispersion parameter in nondrizzling marine stratocumulus, Atmos. Meas. Tech., 6, 1817–1828, https://doi.org/10.5194/amt-6-1817-2013, 2013.

Wu, P., Dong, X., Xi, B., Tian, J., & Ward, D. M. (2020). Profiles of MBL cloud and drizzle microphysical properties retrieved from ground-based observations and validated by aircraft in situ measurements over the Azores. Journal of Geophysical Research: Atmospheres, 125, e2019JD032205. https://doi.org/10.1029/2019JD032205 )

Response: Thank you for the suggestion and the recommended references. An additional brief discussion of ground-based radar's profiling of LWC and ER has been added, as follows:

" Cloud profiles characterized by active radars operated on ground-based sites or spaceborne satellites often served as the truth to validate cloud retrievals from passive sensors (Roebeling et al. 2008). Ground-based radars such as the scanning ARM cloud radars operating at X band (9.4 GHz), Ka band (35 GHz), and W band (94 GHz) are capable to characterize vertical profiles of cloud reflectivities (Kollias et al. 2014; Lhermitte 1988). Combined with liquid water path measured by microwave radiometer and cloud base height identified by Ceilometer, the profiles of LWC, ER and cloud droplet number concentration (CDNC) can be estimated (Dong and Mace 2003; Frisch et al.

1995; Mace and Sassen 2000; Rémillard et al. 2013). It is also reported that ground-based radar could distinguish drizzle from cloud particles (Chen et al. 2008) and derive the LWC and ER profiles of each feature (Wu et al. 2020)."

Line 78: I would recommend repeating the acronym explanation since it is given in the abstract, and a reader interested in the paper might go straight to the text.

Response: revised as suggested.

Line 100: Please specify where these profiles were taken and if the conclusions can be considered general.

Response: we have made a stronger conclusion than it should be. The sentence is revised as: "It is also documented that the LWC profiles are mostly documented as triangular-shaped with a maximum value (turning point) in the middle cloud layer, and the individual cloud nuclei concentration profiles are vertically homogeneous in the middle cloud layer"

Line 105: There's a whole literature on retrieval of cloud and drizzle profiles from ground-based, some works are also mentioned in my previous comments. I think it might deserve a dedicated paragraph, as stated in the general comment and above.

Response: Thank you for stressing out this point. Some studies related to cloud profile retrieval from ground-based instruments are referenced within the additional discussion added in response to the previous comment.

Line 106-108: I think this proposition is quite strong. Validation with other observing platforms is crucial. I am missing a discussion on the limitation of airborne sensors in retrieving quantities like LWP, or LWC for example. What are the possible sources of uncertainty that can affect the retrieval? Also, in Grosvenor there is some discussion on that

Response: The initial text was indeed requiring additional discussion of the possible sources of retrieval uncertainty. The sentence has been revised into:

Even though difficulties exist in the measurements of cloud droplet size profiles from airborne sensors, such as capturing the extremely small or large droplets, unrealistic assumptions and types of probes and their installations, these kinds of datasets are usually considered the truth of the cloud droplet size distribution in simulated retrievals (Alexandrov et al. 2020; Grosvenor et al. 2018).

Line 115: Replace in order to with simply To

Response: revised as suggested.

Line 129: Taking…by an LES: Maybe add a reference that supports this statement.

Response: the following reference is added here.

van der Dussen, J. J., de Roode, S. R., Dal Gesso, S., and Siebesma, A. P.: An LES model study of the influence of the free tropospheric thermodynamic conditions on the stratocumulus response to a climate perturbation, Journal of Advances in Modeling Earth Systems, 7, 670-691, https://doi.org/10.1002/2014MS000380, 2015.

Line 190: Capital letter for we

Response: Revised as suggested.

Line 190-191: by just summing rainwater content prescribed by the LES output over the cloud profile? maybe just complete the sentence with this information for clarity to nonexperts.

Response: Thank you very much for the suggestion, since we have mentioned the integration of the rainwater content, we think the same meaning is conveyed here. The sentence is revised to clarify that the profiles are taken from LES.

Line 249-255: where can I learn how the SP, WP, SE, WE, and ALL subsets are characterized in terms of entrainment and precipitation? sorry if I missed this aspect. Figure 3 shows the distributions but only for all... how is the distribution for the specific subsets? In general, I have some difficulties here in following the discussion because I don't know how to locate in figure 6a/b) the various subsets. For example, in line 252: "among WP profiles, stronger cloud top entrainments correlate to smaller weighting factors for EOF2". How can I see this?

Response: Thank you for asking. For the first two questions, as you mentioned, the SP, WP, SE, WE are exactly characterized from the results shown in Figure 3, the histogram is for all cases while the red lines in Figure 3 is to distinguish the intensity for precipitation and cloud-top entrainment. In this round of revision, we add the description in the caption of Figure 3:

"Histogram of the counts of the rainwater paths (a) and of the cloud top entrainment rates (b) in the RAMS cloud profiles, the red vertical lines from left to right indicate the $25^{th}$, $50^{th}$ and $75^{th}$ percentiles. A profile is defined as strong-precipitating (SP) when the rainwater path exceeds the $75^{th}$ percentile, and weak-precipitating (WP) when the rainwater path remains less than the $25^{th}$ percentile. Similarly, a profile is considered and defined as strong cloud-top entrainment (SE) when the entrainment rate at cloud top exceeds the $75^{th}$ percentile, and weak cloud-top entrainment (WE) when the entrainment rate at cloud top less than the $25^{th}$ percentile."

For the last question, in Figure 6 (a) or (b), we add WE labels aside of current EN<$25^{th}$ label, same changes made to SE, WP and SP. WP corresponds to cases in the third row, in which as the cloud-top entrainment increase (from left to the right) the populated dots are moving downward in y-axis that representing EOF2 weighting factor. Therefore, we say that EOF2 weight factors are getting smaller.

Line 265-266: I find it really hard to follow and understand what you are referring to when you say center bottom boxes and leftmost column. Please try to clarify the description and identify clearly to which boxes you are referring for WE and SE.

Response: Thank you for the suggestion. The referred boxes are underlined in both Figure 7 and Figure 8 to make it easily findable. Then, we pinpoint the location of the boxes in the text as well.

Line 270-271: same as before.

Response: the sentence has been revised as follows:

Examples can be found in the boxes in the top two rows (in Figure (a)-(d)) that corresponding to more pronounced polyline profile that account for in total 28.95% in Figure 7(a) receive 42.01% of samples in Figure 7(b), and in the boxes that account for in total 6.94% in Figure 7(c) receive 13.84% of samples in Figure 7(d).

Line 274-275: Which boxes, again? not clear what are the 16 boxes in the subplots of figure 6. Do you mean the grid 4x4 if visible in each subplot of figure 6B? please, clarify this better, it is very hard to follow.

Response: Sorry for the confusing description. Yes, we meant the grid 4x4 in each subplot of figure 6 (a) and (b). Citing Figure 6 here is to stress out that two binning boundary strategies were adopted. The paragraph is rewritten without citing any other figures, as follows:

As in Section 4.1, the impact of precipitation is analyzed by the fractions of profiles that fall into $4 \times 4$ bins for WP and SP cases (Figure 8). The analyses are performed in two binning methods: the quartile bin boundaries (Figure 8(a) and (b)), and the arithmetic mean bin boundaries without extremes (as in Figure 8(c) and (d)).

Similarly, the starting paragraph for Section 4.1 is rewritten as well in our revised manuscript.

Line 275: I would say: "the most populated region of weighting factors for EO1/EO2 in figure 8 a and b are the left bottom corners"

Response: Thanks for the suggestion which we understand was likely referring to Line 277. The sentence is revised as followed:

Figure 8 indicates that weighting factors for WP cases are populated in the left-bottom corner boxes (Figure 8(a) and (c)), whereas these of SP are populated in the boxes in right two columns (Figure 8(b)) or center-to-right bottom boxes (Figure 8(d)).

Line 276: there's also a 6.83 in the second column at the top that is relevant, in my view. I would re-scale the color bar to make the occurrences more visible. The light blue is too light. Also, the color bar and the numbers are the boxes, aren't they doubling the information?

Response: We do agree with you that 6.83% is not negligible, but it is smaller than the numbers in other boxes where we have underlined with a criteria of 10%. The color and the number are indeed duplicating the information but our intent is to help readers visually interpret the presented numeric values. As identifying the weights from the color only might be difficult especially for close colors, we maintained the color filling and the number inside.

Line 280-282: It is not clear. I would suggest adding a figure where you support the considerations with some graphical representation highlighting what you are summing up.

Response: Thank you for the suggestion. In section 4.1 and 4.2, the changes between SE and WE, SP and WP are showing in the changes of the percentage together with the reconstructed profiles of LWC and ER. An example is the revised Figure 8 as followed; similar changes have been made to Figure 7 to help interpret the results we would like to deliver here.

[Figure]

Figure 8 (a) The percent of profiles for weak precipitation (WP), the bins are characterized by quartile boundaries; (b) Same as (a) but for strong precipitation (SP); (c) The percent of profiles for weak precipitation (WP), the bins are characterized by arithmetic mean boundaries; (d) Same as (c) but for strong precipitation (SP). These boxes that received more than 10% of the examples are underlined in dotted blue line for WE cases and in dotted tangerine for SE cases. These boxes that received more than 10% of the examples are underlined in dotted blue line for WE cases and in dotted tangerine for SE cases. (e) and (f) are the difference of the percent of samples for LWC and ER between SP (b) and WP (a) cases; (j) and (h) are the difference of the percent of samples for LWC and ER between SP (d) and WP (c) cases. In (e)-(h), red color and blue color indicate the increase and decrease of the samples, small variations in percent (within ±3%) are plotted with dotted line.

The changing of the weights in the boxes towards different direction indicates the structural variation as the following figure. Since the x-axis representing the weighting factors for EOF1, which is monotonically increasing, therefore moving to the right means larger slope in the reconstructed profiles. Similarly, y-axis representing the weighting factors for EOF2, which is tringle-shaped, therefore moving to the upper direction means more pounced curved profiles.

[Figure]

Line 350: How do these profiles compare with in situ and ground-based observations? at least a qualitative comment can be interesting

Response: Thank you for the suggestion. We would like to involve this comparison in the future work.

Reference

Alexandrov, M.D., Miller, D.J., Rajapakshe, C., Fridlind, A., van Diedenhoven, B., Cairns, B., Ackerman, A.S., & Zhang, Z. (2020). Vertical profiles of droplet size distributions derived from cloud-side observations by the research scanning polarimeter: Tests on simulated data. Atmospheric Research, 239, 104924

Chen, R., Wood, R., Li, Z., Ferraro, R., & Chang, F.-L. (2008). Studying the vertical variation of cloud droplet effective radius using ship and space-borne remote sensing data. Journal of Geophysical Research, 113

Dong, X., & Mace, G.G. (2003). Profiles of Low-Level Stratus Cloud Microphysics Deduced from Ground-Based Measurements. Journal of Atmospheric and Oceanic Technology, 20, 42-53

Frisch, A.S., Fairall, C.W., & Snider, J.B. (1995). Measurement of Stratus Cloud and Drizzle Parameters in ASTEX with a Kα-Band Doppler Radar and a Microwave Radiometer Journal of Atmospheric Sciences, 52, 2788-2799

Grosvenor, D.P., Sourdeval, O., Zuidema, P., Ackerman, A., Alexandrov, M.D., Bennartz, R., Boers, R., Cairns, B., Chiu, J.C., Christensen, M., Deneke, H., Diamond, M., Feingold, G., Fridlind, A., Hünerbein, A., Knist, C., Kollias, P., Marshak, A., McCoy, D., Merk, D., Painemal, D., Rausch, J., Rosenfeld, D., Russchenberg, H., Seifert, P., Sinclair, K., Stier, P., van Diedenhoven, B., Wendisch, M., Werner, F., Wood, R., Zhang, Z., & Quaas, J. (2018). Remote Sensing of Droplet Number Concentration in Warm Clouds: A Review of the Current State of Knowledge and Perspectives, 56, 409-453

Kollias, P., Bharadwaj, N., Widener, K., Jo, I., & Johnson, K. (2014). Scanning ARM Cloud Radars. Part I: Operational Sampling Strategies. Journal of Atmospheric and Oceanic Technology, 31, 569-582

Lhermitte, R.M. (1988). Cloud and precipitation remote sensing at 94 GHz. Ieee Transactions on Geoscience and Remote Sensing, 26, 207-216

Mace, G.G., & Sassen, K. (2000). A constrained algorithm for retrieval of stratocumulus cloud properties using solar radiation, microwave radiometer, and millimeter cloud radar data, 105, 29099-29108

Rémillard, J., Kollias, P., & Szyrmer, W. (2013). Radar-radiometer retrievals of cloud number

concentration and dispersion parameter in nondrizzling marine stratocumulus. Atmos. Meas. Tech., 6, 1817-1828

Roebeling, R.A., Placidi, S., Donovan, D.P., Russchenberg, H.W.J., & Feijt, A.J. (2008). Validation of liquid cloud property retrievals from SEVIRI using ground-based observations, 35

Wu, P., Dong, X., Xi, B., Tian, J., & Ward, D.M. (2020). Profiles of MBL Cloud and Drizzle Microphysical Properties Retrieved From Ground-Based Observations and Validated by Aircraft In Situ Measurements Over the Azores. Journal of Geophysical Research: Atmospheres, 125, e2019JD032205

---

## Author Comment (AC2)

This study developed an analytical cloud profile model based on the dominant patterns of LWC and ER that derived from simulations of stratocumulus. Cloud profile retrieval from passive satellites is very challenging, this study simplifies the characterization of cloud profiles and enables the potential predication of precipitating or entraining level. The analytical cloud profile model is a very interesting tool for the futuristic retrievals with these main profile patterns are all involved. Overall, this work is very well organized and elaborated, the figures are displayed in a good manner. This work could be accepted for publication after clarifying some minor issues:

1. I noticed that the authors extract the EOFs for ER and LWC simultaneously by grafting the two profiles. The EOFs that derived from this way would be different from extracting ER and LWC separately. I would suggest add discussion on this.

Response: Thank you for the question. The grafting of the two profiles is performed to ensure the EOF1-3 for ER and LWC are representative of the features for the same group of profiles. Actually, we have also done experiments by applying EOF analyses specifically for LWC or ER profiles and the reconstructed dominant structures were also exhibiting triangle shaped features for both LWC and ER profiles. The difference is that the part of variance that the first three EOFs could explain is larger than for the grafted results. However, when interpreting the simultaneous patterns of LWC and ER profiles for a certain level of cloud-top entrainment or precipitation, the individually derived EOFs for LWC and ER could not work. Since introducing the EOF analyses individually for LWC and ER would not help the analyses and the necessities of grafting the profiles are explained in the text, we decided not to bring extra EOF results to the manuscript.

2. The analytical model is based on 4 prominent patterns of LWC and ER profiles that extracted from stratocumulus, does the model works for other liquid clouds?

Response: It is a difficult to answer this question based on our results. The cloud regimes we analyzed in this study remain very typical of stratocumulus even though we tried to simulate a wide range of situations. We can only speculate here that the analytical model is sufficiently generic and will remain valid for the description of other cloud types in terms of its ability to describe a wide range of vertical variation of cloud properties. Further investigations of the LWC-ER patterns relating to even more diverse turbulent or precipitating intensities for other types of liquid clouds may be needed in future work to generalize our results to all types of liquid clouds, especially for the potential links between profile shape, precipitation and entrainment.

3. line 65, In order to reconcile the retrievals performed using different spectral channels some studies assumed that the cloud ER profiles are linear or polylinear with no more than one turning point so that retrieval can be implemented by either a lookup table method. whether the polylinear are triangle shaped as well?

Response: We found the profiles are close to triangle shaped as shown in figure 1. For individual profiles, high-resolution detailed oscillations in the optical thickness axis could be found, but the main triangle shape feature is still distinguishable.

[Figure]

Figure 1. The figure (2) from Chang et al 2002. Vertical profiles of (a) cloud DER, (b) liquid water content, and (c) the dispersion (s) of the lognormal size distribution that were simulated for three cloud geometrical thicknesses of 200, 600, and 1600 m. These profiles were simulated based on the observed values measured at the top, middle, and bottom of a marine stratocumulus cloud during ASTEX.

4. Line 91, the acronym 'LWP' is first introduced here, liquid water content?

Response: Yes, thank you for pointing out. The full name is used instead.

5. Line 105, what do you mean by "with different levels of complexity"

Response: The sentence has been revised as: "LES models with different levels of complexity can capture microphysical processes in response to the effects of turbulent mixing by focusing on different length scales and time scales."

Line 108, probably add more references for airborne measurements of cloud profiles.

Response: Thank you for the suggestion, the description has been detailed with more references and moved close to ground-based profiling measurements. The modified version is as followed:

"Airborne equipped particle probe, imager and spectrometer are able to capture the profile of size distribution and droplet number concentration for cloud and precipitation droplets (Lawson et al.,

2001; Dadashazar et al., 2022) . Even though uncertainties such as capturing the extremely small or large droplets, unrealistic assumptions, types of probes and impact from their installations exist in the measurements, these kinds of datasets provide valuable reference for understanding the cloud profiles in nature and evaluating these simulations or satellite retrievals (Grosvenor et al., 2018; Alexandrov et al., 2020; Zhao et al., 2018)."

Line 109, the ground-based cloud profiling measurements is another choice for validating cloud profile retrievals.

Response: we do agree that ground-based measurements are an important reference for satellite retrievals, the discussion related to ground-based profiling of LWC and ER has been expanded as follows:

[revised manuscript text omitted]

---

## Author Response (AR2)

**General comments:**

Dear Dr. Shang,

Thank you for addressing the concerns by the referees on your manuscript. I am almost ready to accept your manuscript for publication, except for a clarification on the data availability. As the ACP data availability policy states, all data and code needs to be openly available. If this requirement cannot be met, a clear explanation needs to be given. Please update your data availability and/or the statement accordingly. For more information, you can find the statement here: https://www.atmospheric-chemistry-and-physics.net/policies/data_policy.html

Best regards,
Thijs Heus

Response:

Dear Thijs,

Thank you very much for the message. We all agree that the availability of the data and code is an important factor for the publication. All the authors agree to make the datasets (simulations of stratocumulus from RAMS) openly available and the codes to be available upon request from the authors. However, we haven't found an appropriate site to release the datasets since they are quite large. Therefore, in the revised version we further explained that the data are available from the authors and codes are available upon request from the authors. I am not sure if the explanation shall be in an individual file or included in the "Data availability" part in the final version of the paper. Please let me know if some points are missed here. Sorry for the inconvenience!

Best regards,

Huazhe

---

## Author Response (AR3)

Comments: Your "Table 2" contains coloured cells or/and coloured values. Please note that this will not be possible in the final revised version of the paper due to HTML conversion of the paper. When revising the final version, you can use footnotes or italic/bold font. But if the colour spectrum is necessary and cannot be exchanged for footnotes, bold, or italic, then please inform us.

Response: Thank you for the message, the color element or background can be changed into black or gray. The table has been changed as followed. Please let me know if this fit in the on line display.

| | [1] | [2] | [3] | [4] |
|---|---|---|---|---|
| ALL | *60.09%* | *25.45%* | *7.63%* | *6.82%* |
| WE | 81.55% | 11.66% | 4.49% | 2.31% |
| SE | 42.44% | 38.99% | 9.97% | 8.61% |
| WP | 52.20% | 27.81% | 14.65% | 5.34% |
| SP | 61.73% | 26.71% | 3.04% | 8.52% |